# Knowledge, attitude, and practice toward genetic testing in breast cancer patients in China

Xin Ye[1,2,3], Ting Xu[1,2,3], Jun Cao[4], Gang Li[5], Ming Zhuang[6], Guangfu Hu[7], Hongliang Chen[8], Min Wang[9], Jie Wang[1,2,3]*

1 Department of Breast Surgery, The International Peace Maternity and Child Health Hospital, School of Medicine, Shanghai Jiao Tong University, Shanghai, China, 2 Shanghai Key Laboratory of Embryo Original Diseases, Shanghai, China, 3 Shanghai Municipal Key Clinical Specialty, Shanghai, China, 4 Department of Breast and Urologic Medical Oncology, Fudan University Shanghai Cancer Center, Shanghai, China, 5 Medical Oncology, Minhang Branch, Fudan University Shanghai Cancer Center, Shanghai, China, 6 Breast Surgery Department, Xinhua Hospital Affiliated to Shanghai Jiao Tong University School of Medicine, Shanghai, China, 7 Department of Breast Surgery, Huangpu Branch, Shanghai Ninth People's Hospital, Affiliated to Shanghai Jiao Tong University School of Medicine, Shanghai, China, 8 Department of Breast Surgery, Obstetrics and Gynecology Hospital of Fudan University, Shanghai, China, 9 Breast-Thyroid Department, Shanghai General Hospital, Shanghai Jiao Tong University School of Medicine, Shanghai, China

☯ These authors contributed equally to this work.
* jiewang76@hotmail.com

## Abstract

Genetic testing is widely recommended in the diagnosis and management of breast cancer. This study aimed to investigate the knowledge, attitude, and practice (KAP) toward genetic testing in Chinese patients with breast cancer. This multicenter cross-sectional study enrolled breast cancer patients in seven public hospitals in Shanghai, China, between November 2022 and January 2023. A self-administered web-based questionnaire was used to collect the participants' demographic information and their KAP regarding genetic testing. A total of 592 valid questionnaires were collected in this study; 145 (24.49%) patients underwent genetic testing, and 20.61% of the patients never learned about genetic testing. The knowledge, attitude, and practice scores were 4.59±4.49 (22.95%, possible range: 0–20), 16.72±2.92 (83.60%, possible range: 0–20), and 23.67±5.18 (73.97%, possible range: 0–32), respectively. Multivariable logistic regression showed that knowledge (OR=1.21, 95%CI: 1.51–1.28, $P<0.001$), attitude (OR=1.10, 95%CI: 1.01–1.19, $P=0.027$), Jiangsu Province (OR=0.40, 95%CI: 0.19–0.84, $P=0.016$), monthly income of 5000–10,000 CNY (OR=0.46, 95%CI: 0.25–0.86, $P=0.015$), disease duration of 5–10 years (OR=0.50, 95%CI: 0.27–0.94, $P=0.030$) and disease duration of ≥10 years (OR=0.26, 95%CI: 0.11–0.60, $P=0.002$), triple-negative subtype (OR=3.45, 95%CI: 1.51–7.85, $P=0.003$) were independently associated with patients' behavior of undergoing genetic testing. The structural equation modeling showed that knowledge directly positively influenced attitude (β=0.343, $P<0.001$), while attitude directly positively influenced practice (β=0.942, $P<0.001$). Chinese patients with

**Data availability statement:** All relevant data are within the manuscript and its Supporting Information files.

**Funding:** This work was supported by the Shanghai Developing Center of Shenkang Hospital (SHDC22020209) and the Shanghai Municipal Key Clinical Specialty (shsl-czdzk06302). The funders had no role in study design, data collection and analysis, decision to publish, or preparation of the manuscript.

**Competing interests:** The authors have declared that no competing interests exist.

breast cancer demonstrated poor knowledge, positive attitude, and suboptimal practice toward genetic testing. More education and counseling on genetic testing for patients are necessary.

## Introduction

According to GLOBOCAN 2020, breast cancer has become the most commonly diagnosed cancer in women [1]. In China, there were an estimated 429,105 new cases of breast cancer [1–3], raising attention to the prevention, diagnosis, and management of breast cancer [4–6]. Mutations in the BRCA1 and BRCA2 genes could explain about 5%-10% of breast cancers [7], and other genes (e.g., TP53, CHEK2, and ATM) are also involved in breast cancer risk when mutated [8]. Genetic testing can profoundly influence the management of breast cancer. Indeed, patients carrying BRCA1 mutations have a lifetime risk of breast cancer of 40%–87%, while the risk in those with BRCA2 mutations is 25%–30% [9,10]. Patients carrying such mutations are eligible for prophylactic bilateral mastectomy [11] or contralateral prophylactic mastectomy when undergoing surgery for breast cancer [12]. In addition, a proven mutation warrants consideration of more intensive screening or follow-up for cancer new onset or recurrence [10,13]. BRCA1 and BRCA2 carriers are also eligible for adjuvant treatments with PARP inhibitors, such as olaparib [14]. Patients with mutations in TP53 are at increased risk of cancers, but they are also particularly sensitive to radiation, and their management should be done while avoiding X-rays [15]. Hence, genetic testing has become an established part of breast cancer management [16–18], but public knowledge regarding genetic risk appears suboptimal [19–21].

A knowledge, attitude, and practice (KAP) survey is a widely used cross-sectional method that provides real-world data for medical strategies and clinical practice [22]. Previous studies explored the KAP of genetic testing for breast cancer among patients in the United States of America (USA) and college students in China [23,24]. However, few studies analyzed the relationship between KAP among patients with breast cancer regarding genetic testing in China. Understanding the benefits, limitations, and potential consequences of genetic testing can empower patients to make informed choices about genetic testing. Patients with positive genetic test results might take proactive steps to manage their risk and improve outcomes.

Therefore, this study aimed to investigate the KAP towards genetic testing in patients with breast cancer in China. This will help explore the reasons for the low rate of genetic testing, improve methods of genetic testing, and better assist and guide clinical diagnosis, follow-up, and drug application.

## Methods

### Study design and participants

This multicenter cross-sectional study was conducted between November 2022 and January 2023 and included breast cancer patients from seven public hospitals in Shanghai, China (the International Peace Maternity and Child Health Hospital, School of Medicine, Shanghai Jiao Tong University; Fudan University Shanghai Cancer

Center; Huangpu Branch, Shanghai Ninth People's Hospital, Affiliated to Shanghai Jiao Tong University School of Medicine; Xin Hua Hospital Affiliated to Shanghai Jiao Tong University School of Medicine; Shanghai General Hospital, Shanghai Jiao Tong University School of medicine; Minhang Branch, Fudan University Shanghai Cancer Center; Obstetrics and Gynecology Hospital of Fudan University).

The inclusion criteria were patients with breast cancer and expected survival of more than 1 month. The exclusion criteria were: 1) delirious, 2) loss of consciousness, or 3) unable to read or complete the questionnaire for any reason. This study was approved by the International Peace Maternity and Child Health Hospital, School of Medicine, Shanghai Jiao Tong University (GKLW2022–51), and written informed consent was obtained from the participants.

## Questionnaire

The questionnaire was designed with reference to the Breast Cancer Screening Guidelines for Chinese Women [25], the Chinese Guidelines for Breast Cancer Screening and Early Diagnosis [26], the Expert Consensus on BRCA1/2 Gene Testing and Clinical Application in Chinese Breast Cancer Patients (2018) [27], and the Consensus Guidelines on Genetic Testing for Hereditary Breast Cancer from the American Society of Breast Surgeons [28]. The draft was pretested (n = 68) with a Cronbach's α of 0.74, demonstrating good internal consistency.

The final questionnaire was in Chinese, with a total of 43 items. There were 15 items about demographic information, 10 about knowledge, 7 about attitude, and 11 about practice. For the knowledge items, statements with related knowledge "well informed", "partially informed", and "unknown" were scored 2, 1, and 0, with a possible score range of 0–20 points. Attitude was scored on a scale from highly positive (4 points) to extremely negative (0 points). Items A1 and A2 did not involve a clear positive or negative tendency, and descriptive statistics were conducted. The possible total score range of attitude was 0–20. Practice was scored on a scale of 0–4 points based on their positivity, while items P1, P4-1, and P10 could not be assigned scores. Item P1 was used to determine the participants' behavior in undergoing genetic testing; item P4-1 was optional, and item P10 focused on the channels. Thus, the possible total score range of practice was 0–32. Total scores ≥ 80% were considered good knowledge, positive attitude, and proactive practice.

Data collection was conducted using the *Sojump* platform (www.sojump.com), which allows for online survey distribution and collection. The study participants were recruited through consultation rooms and social network platforms such as WeChat groups. The questionnaires were completed anonymously to protect the privacy of the participants. In addition, to ensure that the data was not skewed by repeat submissions, the platform only allowed one submission per IP address. For data quality, the questionnaires with the same option selected for all items were excluded from the analysis.

## Sample size

The formula

$$n = \left(\frac{Z_{1-\alpha/2}}{\delta}\right)^2 \times p \times (1-p)$$

was used to calculate the sample size of cross-sectional surveys. In the formula, $n$ represents the sample size for each group, $a$ represents the type I error (which is typically set at 0.05), $Z_{1-\alpha/2} = 1.96$, $\delta$ represents the allowable error (typically set at 0.05), and $p$ is set at 0.5 (as setting it at 0.5 maximizes the value and ensures a sufficiently large sample size). Hence, the calculated sample size was 384. Considering an estimated questionnaire response rate of 80%, a minimum of 480 valid questionnaires were needed.

## Statistical analysis

Stata 17.0 (Stata Corporation, College Station, TX, USA) was used for analysis. The continuous variables were presented as means ± standard deviations (SD), and independent-samples t-test or one-way analysis of variance (ANOVA) was used

for analysis. The categorical variables were presented as n (%). Pearson correlation was used to analyze the correlation between pairs of knowledge, attitude, and practice scores. The variables that were statistically significant in univariable logistic regression analyses were included in the multivariable logistic regression analysis. Multivariable logistic regression was conducted to determine the factors associated with patients' behavior of undergoing genetic testing (item P1). A structural equation modeling (SEM) analysis was used to evaluate the hypothesis that knowledge might directly affect attitude and practice, as well as indirectly influence practice through attitude. Two-sided P-values < 0.05 were considered statistically significant.

## Results

A total of 597 questionnaires were collected, and five were excluded due to contradictory logic or patterned duplicate answers, resulting in 592 valid questionnaires (99.16%). Most participants were 41–60 years old (427, 72.13%) and married (519, 87.67%) (Table 1).

The mean knowledge score was 4.59 ± 4.49 (22.95%, possible range: 0–20). The results showed differences in knowledge across age, marital status, residence, work status, income, and molecular subtype (all $P < 0.05$). There were 224 (37.84%) patients who did not know about the knowledge in each item except for K1, "What genetic is testing for breast cancer" (Table 2). The attitude score was 16.72 ± 2.92 (83.60%, possible range: 0–20). The attitude score varied with age, marital status, maternity situation (including pregnancy and childbirth), education, and work status (all $P < 0.05$). About one-third of the patients maintained a neutral view on most questions (Table 3). The mean score of practice was 23.67 ± 5.18 (73.97%, possible range: 0–32). The results showed the differences in the practice score in different ages, marital statuses, provinces, education, working status, disease duration, and molecular subtype (all $P < 0.05$). Most participants (93.76%) were willing to follow the doctors' instructions and follow-up examination, while there were still some participants who remained equivocal, referring to prophylactic mastectomy and desire to learn about genetic testing (Table 4).

Most participants had never undergone genetic testing for breast cancer (75.51%) (P1). The main access to knowledge about breast cancer and genetic testing was community preaching and communication between friends, accounting for 33.28% and 28.4%. There were 20.61% of the participants never learned about related information on breast cancer and genetic testing. Complete unawareness, economic considerations, and personal psychological factors were the reasons that influenced the patients to remind their children and other close family members to undergo breast cancer genetic testing (Fig 1).

Knowledge (r = 0.348, $P < 0.001$) and attitude (r = 0.464, $P < 0.001$) were positively related to the practice of patients with breast cancer regarding genetic testing (Table 5). The multivariable logistic regression showed that knowledge (OR = 1.21, 95%CI: 1.51–1.28, $P < 0.001$), attitude (OR = 1.10, 95%CI: 1.01–1.19, $P = 0.027$), Jiangsu Province (OR = 0.40, 95%CI: 0.19–0.84, $P = 0.016$), monthly income of 5000–10,000 ¥/CNY (OR = 0.46, 95%CI: 0.25–0.86, $P = 0.015$), disease duration of 5–10 years (OR = 0.50, 95%CI: 0.27–0.94, $P = 0.030$), disease duration of ≥ 10 years (OR = 0.26, 95%CI: 0.11–0.60, $P = 0.002$), and triple-negative breast cancer (OR = 3.45, 95%CI: 1.51–7.85, $P = 0.003$) were independently associated with patients' behavior of undergoing genetic testing (item P1) (Table 6).

The SEM analysis (Fig 2) showed that knowledge directly positively influenced attitude (β = 0.343, $P < 0.001$) but not practice (β = 0.036, $P = 0.528$), while attitude directly positively influenced practice (β = 0.942, $P < 0.001$) (S1 Table). Two of the five model fit indices indicated good fitting (S2 Table).

## Discussion

This study demonstrated poor knowledge, positive attitude, and suboptimal practice towards genetic testing among Chinese patients with breast cancer. It is crucial to improve patients' understanding of genetic testing for better practice.

The patients in this study demonstrated poor knowledge, which was consistent with previous studies that also found that knowledge of breast cancer patients in Southeast Texas and Georgia about genetic testing was poor, while most of

**Table 1. Demographic characteristics.**

| Variables | n (%) | Knowledge score | | Attitude score | | Practice score | |
|---|---|---|---|---|---|---|---|
| | | Mean±SD | P | Mean±SD | P | Mean±SD | P |
| **Total** | 592 (100) | 4.56±4.49 | | 16.72±2.92 | | 23.67±5.18 | |
| **Age, years** | | | 0.006 | | <0.001 | | <0.001 |
| ≤ 40 | 91 (15.50) | 6.10±5.62 | | 17.99±2.97 | | 25.53±5.31 | |
| 41-50 | 248 (42.25) | 4.73±4.50 | | 16.74±2.92 | | 23.94±5.07 | |
| 51-60 | 179 (30.49) | 3.60±3.81 | | 16.28±2.75 | | 23.03±5.18 | |
| ≥ 61 | 69 (11.75) | 4.51±3.85 | | 16.16±2.75 | | 21.91±4.75 | |
| **Marital status** | | | 0.021 | | 0.031 | | 0.005 |
| Unmarried | 14 (2.36) | 7.00±6.49 | | 16.50±3.37 | | 25.36±3.27 | |
| Married | 519 (87.67) | 4.66±4.50 | | 16.81±2.89 | | 23.83±5.18 | |
| Divorced/Widowed | 59 (9.97) | 3.10±3.37 | | 15.98±3.01 | | 21.81±5.27 | |
| **Ethnicity** | | | 0.184 | | 0.653 | | 0.673 |
| Han | 577 (97.47) | 4.60±4.50 | | 16.73±2.93 | | 23.70±5.16 | |
| Others | 15 (2.53) | 3.00±3.76 | | 16.33±2.41 | | 22.47±6.02 | |
| **Reproductive history** | | | | | | | |
| Pregnancy (times) | | | 0.924 | | 0.036 | | 0.823 |
| 0 | 44 (7.48) | 4.89±5.33 | | 16.09±3.37 | | 23.75±5.00 | |
| 1 | 207 (35.20) | 4.77±4.70 | | 16.46±2.78 | | 23.59±5.39 | |
| ≥ 2 | 337 (57.31) | 4.42±4.24 | | 16.99±2.92 | | 23.73±5.09 | |
| Childbirth (times) | | | 0.551 | | 0.018 | | 0.127 |
| 0 | 46 (7.80) | 4.98±5.39 | | 16.07±3.17 | | 23.83±5.55 | |
| 1 | 431 (73.05) | 4.62±4.42 | | 16.61±2.89 | | 23.44±5.20 | |
| ≥ 2 | 113 (19.15) | 4.18±4.39 | | 17.35±2.83 | | 24.35±4.94 | |
| **Province** | | | 0.217 | | 0.210 | | 0.046 |
| Jiangsu | 118 (19.93) | 4.68±4.19 | | 17.02±2.64 | | 23.85±4.19 | |
| Zhejiang | 33 (5.57) | 5.58±4.23 | | 17.15±2.75 | | 25.67±4.96 | |
| Shanghai | 247 (41.72) | 4.61±4.63 | | 16.46±2.98 | | 23.23±5.26 | |
| Others | 194 (32.77) | 4.25±4.54 | | 16.80±3.02 | | 23.77±5.60 | |
| **Residence** | | | 0.005 | | 0.417 | | 0.773 |
| Urban | 431 (72.80) | 4.86±4.56 | | 16.64±2.88 | | 23.64±5.34 | |
| Non - urban | 161 (27.20) | 3.76±4.20 | | 16.94±3.03 | | 23.73±4.76 | |
| **Education** | | | < 0.001 | | 0.017 | | 0.021 |
| Middle School and below | 130 (21.96) | 3.63±3.98 | | 16.18±3.07 | | 23.23±4.86 | |
| High School/ Technical secondary school | 157 (26.52) | 3.71±3.91 | | 16.52±2.69 | | 22.96±4.70 | |
| Junior College/ Bachelor's degree and above | 305 (51.52) | 5.39±4.82 | | 17.06±2.93 | | 24.21±5.50 | |
| **Employment status** | | | 0.025 | | 0.030 | | 0.005 |
| Employed | 258 (43.58) | 5.29±5.01 | | 16.99±2.91 | | 24.26±5.23 | |
| Retirement | 212 (35.81) | 4.00±3.96 | | 16.29±2.74 | | 22.67±5.18 | |
| Others | 122 (20.61) | 4.00±3.97 | | 16.90±3.16 | | 24.13±4.86 | |
| **Monthly income, CNY** | | | < 0.001 | | 0.760 | | 0.431 |
| <5000 | 227 (38.34) | 3.71±3.90 | | 16.60±3.02 | | 23.40±5.11 | |
| 5000–10000 | 169 (28.55) | 4.70±4.72 | | 16.82±2.89 | | 23.67±5.14 | |
| >10000 | 196 (33.11) | 5.41±4.77 | | 16.77±2.83 | | 23.97±5.31 | |
| **Medical insurance** | | | 0.888 | | 0.349 | | 0.727 |
| With medical insurance | 588 (99.32) | 4.55±4.47 | | 16.71±2.92 | | 23.66±5.19 | |
| Without medical insurance | 4 (0.68) | 6.25±7.32 | | 18.00±2.16 | | 24.50±5.20 | |

*(Continued)*

**Table 1.** (Continued)

| Variables | n (%) | Knowledge score | | Attitude score | | Practice score | |
|---|---|---|---|---|---|---|---|
| | | Mean±SD | P | Mean±SD | P | Mean±SD | P |
| **Disease duration, years** | | | 0.264 | | 0.248 | | 0.001 |
| < 2 | 132 (22.30) | 4.90±4.77 | | 17.11±2.96 | | 24.80±5.27 | |
| [2, 5) | 179 (30.24) | 5.06±4.87 | | 16.77±2.78 | | 24.20±4.78 | |
| [5, 10) | 184 (31.08) | 4.25±4.10 | | 16.59±2.98 | | 22.84±5.40 | |
| ≥ 10 | 97 (16.39) | 3.76±3.97 | | 16.36±2.98 | | 22.71±5.01 | |
| **Molecular subtype** | | | < 0.001 | | 0.235 | | 0.001 |
| Luminal A | 104 (17.57) | 4.62±4.41 | | 16.92±3.09 | | 24.21±5.30 | |
| Luminal B | 106 (17.91) | 6.36±4.94 | | 16.85±2.94 | | 24.45±4.83 | |
| HER-2 overexpressing | 109 (18.41) | 4.53±4.01 | | 16.81±2.78 | | 23.92±4.68 | |
| Triple-negative | 64 (10.81) | 6.58±5.48 | | 17.25±2.85 | | 24.80±5.58 | |
| Unclear | 209 (35.30) | 3.01±3.56 | | 16.35±2.90 | | 22.52±5.26 | |
| **Treatments** (multiple choice) | | | – | | – | | – |
| Chemotherapy | 508 (85.81) | 4.65±4.50 | | 16.83±2.93 | | 23.78±5.17 | |
| Radiotherapy | 333 (56.25) | 4.96±4.73 | | 16.77±2.91 | | 23.96±4.89 | |
| Endocrine therapy | 407 (68.75) | 4.49±4.39 | | 16.71±2.96 | | 23.51±5.05 | |
| Targeted therapy | 149 (25.17) | 4.64±4.43 | | 16.94±2.80 | | 24.32±4.95 | |
| Surgery | 384 (64.86) | 4.54±4.43 | | 16.98±2.91 | | 23.68±5.09 | |
| Traditional Chinese medicine | 153 (25.84) | 5.12±4.52 | | 16.79±2.74 | | 23.94±4.79 | |
| Others | 15 (2.53) | 7.73±6.47 | | 16.73±2.99 | | 26.40±6.66 | |
| **Comorbidity** | | | 0.798 | | 0.404 | | 0.737 |
| None | 506 (85.47) | 4.61±4.56 | | 16.68±2.92 | | 23.70±5.23 | |
| Ovarian cancer | 3 (0.51) | 6.33±5.69 | | 19.33±4.04 | | 25.33±7.02 | |
| Others | 83 (14.02) | 4.20±4.01 | | 16.87±2.86 | | 23.42±4.88 | |
| **Family history** (multiple choice) | | | – | | – | | – |
| Breast cancer | 100 (16.89) | 5.30±4.66 | | 16.51±2.90 | | 23.44±5.24 | |
| Ovarian cancer | 21 (3.55) | 6.00±6.36 | | 17.71±3.48 | | 24.62±6.69 | |
| Pancreatic cancer | 21 (3.55) | 4.24±5.13 | | 16.62±3.32 | | 23.62±6.06 | |
| Prostate Cancer | 10 (1.69) | 5.60±6.13 | | 17.70±2.11 | | 25.70±5.06 | |

them showed a positive attitude [29,30]. This discrepancy between knowledge and attitude can be due to the trust of the patients that their physicians will provide them with good advice and recommendations, even if they do not understand the rationale behind the advice and recommendations. The study found that education and income were positively related to knowledge. Higher socioeconomic status is generally considered related to better health literacy [31], as also observed in the present study. Another study including newly diagnosed breast cancer participants in the iCanDecide trial also reported that the knowledge concerning genetic testing about breast cancer remained low and emphasized education and propaganda of gene technology [32]. The present study strongly suggests that it is urgent to emphasize the importance and necessity of genetic testing in cancer management, leading to a more systematic and comprehensive understanding of genetic testing in China. There are several advantages to genetic testing. Indeed, the actual risk of new cancer or cancer recurrence of patients carrying the BRCA1/2 mutation gene can be quantified, which allows personalized management. Patients carrying BRCA1/2 mutations have an increased risk of ovarian cancer, necessitating close monitoring and follow-up of the ovaries and uterus during the follow-up process. Other family members can also be informed and choose

**Table 2. Knowledge.**

| Items, n (%) | Have known | Know a part | Don't know |
|---|---|---|---|
| K1. What is genetic testing for breast cancer? | 86 (14.53) | 282 (47.64) | 224 (37.84) |
| K2. Do you know about the clinics of genetic counseling in oncology? | 34 (5.74) | 155 (26.18) | 403 (68.07) |
| K3. 5%~10% of breast cancer patients have clear genetic gene mutations called hereditary breast cancer. BRCA1/2 gene mutations account for 15%, and other related risk genes include TP53, CDH1, PTEN, CHEK2, ATM, and PALB2. | 27 (4.56) | 205 (34.63) | 360 (60.81) |
| K4. The applicable population for breast cancer genetic testing. | 41 (6.93) | 209 (35.30) | 342 (57.77) |
| K5. BRCA1 and BRCA2 are tumor suppressor genes; compared with the general population, BRCA1/2 gene mutation carriers have an increased risk of breast cancer. | 54 (9.12) | 161 (27.20) | 377 (63.68) |
| K6. Women with BRCA1/2 mutations are at increased risk of developing not only breast cancer but also other cancers such as ovarian cancer, fallopian tube cancer, pancreatic cancer, gastrointestinal cancer, and melanoma. | 41 (6.93) | 160 (27.03) | 391 (66.05) |
| K7. For breast cancer patients with BRCA1/2 gene mutation, there is a relatively elevated risk of recurrence after surgery, so a postoperative MRI should be done. | 35 (5.91) | 149 (25.17) | 408 (68.92) |
| K8. Genetic testing for the BRCA1/2 gene can guide the chemotherapy regimens and the targeted therapies. | 49 (8.28) | 173 (29.22) | 370 (62.50) |
| K9. The high sensitivity of breast MRI is helpful for the early detection of small lesions; For women with a high risk of breast cancer, such as those carrying BRCA1/2 and TP53 mutations, regular breast MRI is recommended. | 48 (8.11) | 177 (29.90) | 367 (61.99) |
| K10. Breast cancer patients with hereditary gene mutations should seek the help of doctors before pregnancy, such as embryo screening through assisted reproductive technology to prevent the inheritance of mutant genes to the next generation. | 33 (5.57) | 132 (22.30) | 427 (72.13) |

to undergo genetic testing, also allowing personalized management to screen for breast cancer. Prophylactic measures can be taken in individuals at high risk of cancer, including breast cancer and ovary cancer, and management and follow-up can be adjusted if there is a high risk of recurrence after cancer treatments. The patients can become eligible for effective treatments (e.g., olaparib in patients with mutations in BRCA1 or BRCA2). On the other hand, genetic testing can be associated with stress, anxiety, fear, and exclusion from life or health insurance. It is the physician's role to provide comprehensive information to the patient and help her make an informed decision after balancing the benefits and risks [10,33].

Another study compared the knowledge of breast cancer-related genes between Chinese American and non-Hispanic white patients, and the results showed that Chinese Americans had less knowledge of genetic testing than non-Hispanic white participants [24], which may support the results of this study. Nevertheless, another study reported that although there were differences in knowledge between Black and White women, the knowledge about genetic testing was both high [34]. Of course, different populations in different countries will lead to differences in knowledge due to discrepancies in socioeconomic status and health literacy. Previous studies suggested that practice could be determined by knowledge and attitude [20,35]. This study also illustrated the positive correlation between knowledge and attitude with practice. The SEM analysis also showed that knowledge positively influenced attitude, which in turn positively influenced practice. The results also indicated that shorter disease duration, higher income, and triple-negative subtype were independently associated with patients' behavior of undergoing genetic testing. The present study was cross-sectional, and causality cannot be inferred. The association between shorter disease duration and genetic testing may stem from patients' anxiety about

**Table 3. Attitude.**

| Items | Extremely (agree), n (%) | Relatively (agree), n (%) | Moderately/ Neutral, n (%) | Relatively not (agree), n (%) | Extremely not (agree), n (%) |
|---|---|---|---|---|---|
| A1. Do you feel anxious after being diagnosed with breast cancer? | 179 (30.24) | 209 (35.30) | 148 (25.00) | 43 (7.26) | 13 (2.20) |
| A2. How worried would you be about the recurrence of breast cancer after surgery, assuming you have been treated for it? | 143 (24.16) | 221 (37.33) | 185 (31.25) | 34 (5.74) | 9 (1.52) |
| A3. Are you worried that breast cancer will be passed on to your child? | 137 (23.14) | 185 (31.25) | 134 (22.64) | 114 (19.26) | 22 (3.72) |
| A4. Your willingness for breast cancer genetic testing. | 165 (27.87) | 176 (29.73) | 223 (37.67) | 22 (3.72) | 6 (1.01) |
| A5. In your opinion, if a breast cancer patient has undergone a bilateral mastectomy, there is no need to carry out genetic testing for breast cancer. | 32 (5.41) | 63 (10.64) | 230 (38.85) | 226 (38.18) | 41 (6.93) |
| A6. Patients will be more worried about recurrence if they receive a positive result indicating a pathogenic genetic mutation, so it is better not to have genetic testing. | 21 (3.55) | 83 (14.02) | 186 (31.42) | 256 (43.24) | 46 (7.77) |
| A7. If a genetic mutation with pathogenicity is found by genetic testing, there will be an increase in the frequency of examinations and tests, as well as the associated costs; Are you willing to cooperate? | 140 (23.65) | 231 (39.02) | 191 (32.26) | 26 (4.39) | 4 (0.68) |

* Items A5 and A6 were scored in reverse.

**Table 4. Practice.**

| Items | Definitely yes, n (%) | Probably yes, n (%) | Maybe yes, n (%) | Probably not, n (%) | Definitely not, n (%) |
|---|---|---|---|---|---|
| P2. You follow your doctor's instructions and go to the hospital regularly. | 387 (65.37) | 168 (28.38) | 32 (5.41) | 4 (0.68) | 1 (0.17) |
| P3. If you are found to carry the genetic mutation by genetic testing, will you let your child undergo breast cancer genetic testing? | 260 (43.92) | 192 (32.43) | 105 (17.74) | 23 (3.89) | 12 (2.03) |
| P4. If you are found to carry the genetic mutation by genetic testing, will you remind your other close relatives, other than your child, to undergo breast cancer genetic testing? | 221 (37.33) | 192 (32.43) | 129 (21.79) | 42 (7.09) | 8 (1.35) |
| P5. If you are found to carry the genetic mutation by genetic testing, will you pay attention to other diseases associated with this gene? For example, if you are found to carry the BRCA1/2 gene mutation, will you also pay attention to ovarian cancer, etc., and have regular relevant check-ups? | 264 (44.59) | 230 (38.85) | 82 (13.85) | 11 (1.86) | 5 (0.84) |
| P6. If you are found to carry the genetic mutation by genetic testing, would you consider prophylactic mastectomy? (Such as contralateral prophylactic mastectomy and prophylactic oophorectomy). | 84 (14.19) | 156 (26.35) | 178 (30.07) | 156 (26.35) | 18 (3.04) |
| P7. If you are found to carry the genetic mutation by genetic testing, you will consider receiving gene-targeting medication. | 154 (26.01) | 239 (40.37) | 148 (25.00) | 43 (7.26) | 8 (1.35) |
| P8. If you are found to carry the genetic mutation by genetic testing, and you have fertility needs, will you consider assisted reproductive technology for embryo screening? | 105 (17.74) | 182 (30.74) | 130 (21.96) | 126 (21.28) | 49 (8.28) |
| P9. You are proactive in learning about breast cancer and genetic testing. | 64 (10.81) | 178 (30.07) | 306 (51.69) | 38 (6.42) | 6 (1.01) |

the risk of recurrence or the potential for breast cancer in their children. Additionally, this relationship could be influenced by the presence of more aggressive disease or genetic risk factors, which may lead physicians to recommend genetic testing. Additional studies are warranted regarding that point. The association between practice and triple-negative breast cancer is probably due to the influence of physicians strongly recommending genetic testing in the

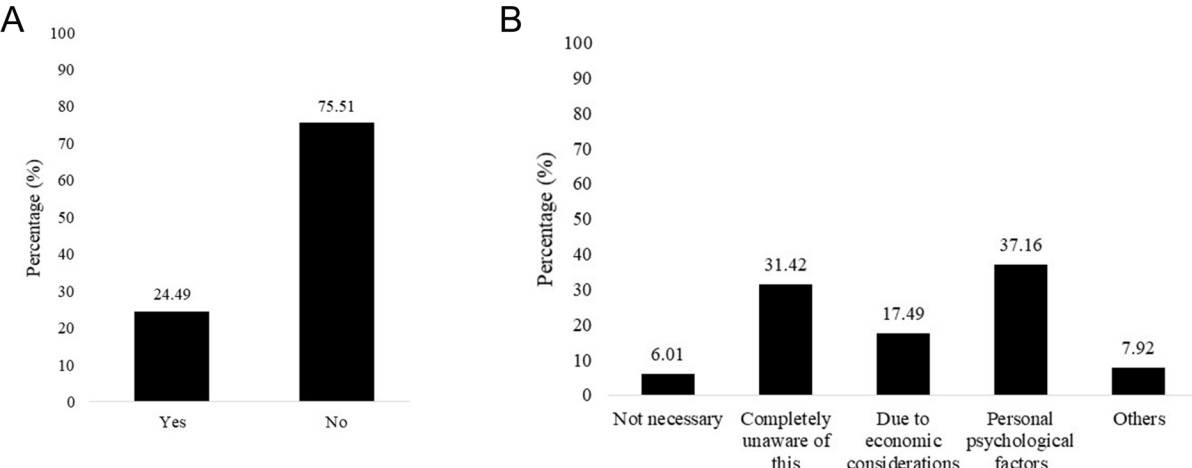

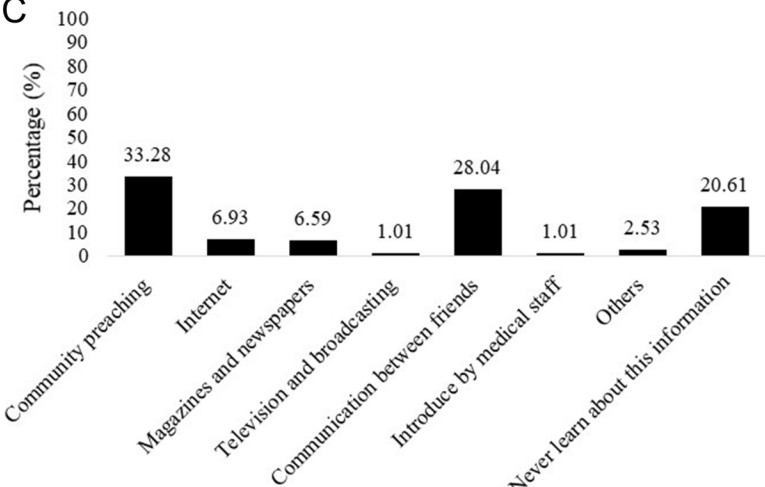

**Fig 1.** (A) P1: have you ever undergone genetic testing for breast cancer or not? (B) P4-1: reasons that influence patients to remind their children and other close family members to undergo breast cancer genetic testing. (C) P10: access to knowledge about breast cancer and genetic testing.

**Table 5. Correlation.**

|  | Knowledge | Attitude | Practice |
|---|---|---|---|
| Knowledge | 1 |  |  |
| Attitude | 0.138 (P<0.001) | 1 |  |
| Practice | 0.348 (P<0.001) | 0.464 (P<0.001) | 1 |

presence of triple-negative breast cancer [36]. Indeed, such cancers are over-represented among BRCA1 and BRCA2 carriers [37,38]. On the other hand, the situation of genetic testing for breast cancer is not optimistic [39–41], in line with the results in this study (only 24.49% had undergone genetic testing). Unfortunately, the present study was not designed to determine the reasons for the low testing rate. Nevertheless, possible reasons include the costs, the physician's KAP toward genetic testing, and the fear of being tested positive. Future studies should examine those reasons.

**Table 6. Multivariable logistic regression analysis.**

| Variables | OR (95%CI) | P |
|---|---|---|
| Knowledge score | 1.21 (1.15, 1.28) | < 0.001 |
| Attitude score | 1.10 (1.01, 1.19) | 0.027 |
| Age, years | | |
| ≤ 40 | Ref. | |
| 41-50 | 0.95 (0.49, 1.85) | 0.889 |
| 51-60 | 0.98 (0.46, 2.06) | 0.949 |
| ≥ 61 | 0.94 (0.36, 2.42) | 0.891 |
| Pregnancy, times | | |
| 0 | Ref. | |
| 1 | 0.65 (0.16, 2.69) | 0.555 |
| ≥2 | 0.57 (0.14, 2.35) | 0.437 |
| Childbirth, times | | |
| 0 | Ref. | |
| 1 | 0.53 (0.13, 2.15) | 0.374 |
| ≥ 2 | 0.37 (0.08, 1.70) | 0.203 |
| Province | | |
| Shanghai | Ref. | |
| Jiangsu | 0.40 (0.19, 0.84) | 0.016 |
| Zhejiang | 0.31 (0.88, 6.06) | 0.090 |
| Others | 1.61 (0.90, 2.86) | 0.109 |
| Education | | |
| Middle School and below | Ref. | |
| High School/Technical secondary school | 1.72 (0.87, 3.41) | 0.121 |
| Junior College/ Bachelor's degree and above | 1.81 (0.90, 3.65) | 0.099 |
| Monthly income, CNY | | |
| <5000 | Ref. | |
| 5000-10,000 | 0.46 (0.25, 0.86) | 0.015 |
| >10,000 | 1.00 (0.56, 1.80) | 0.995 |
| Disease duration, years | | |
| < 2 | Ref. | |
| [2, 5) | 0.90 (0.50, 1.62) | 0.719 |
| [5, 10) | 0.50 (0.27, 0.94) | 0.030 |
| ≥ 10 | 0.26 (0.11, 0.60) | 0.002 |
| Molecular subtype | | |
| Luminal A | Ref. | |
| Luminal B | 1.53 (0.73, 3.20) | 0.263 |
| HER-2 overexpressing | 1.95 (0.91, 4.18) | 0.084 |
| Triple-negative | 3.45 (1.51, 7.85) | 0.003 |
| Unclear | 1.79 (0.88, 3.63) | 0.107 |

Patients <40 years old could be more likely to know of genetic testing because genetics in oncology is a young discipline with which older patients can be less familiar. Genetics is a topic that has been covered in secondary school in China for a little more than two decades, making younger patients more likely to be knowledgeable about genetics. Young patients are also more likely to access the internet frequently to seek knowledge than older patients. Furthermore,

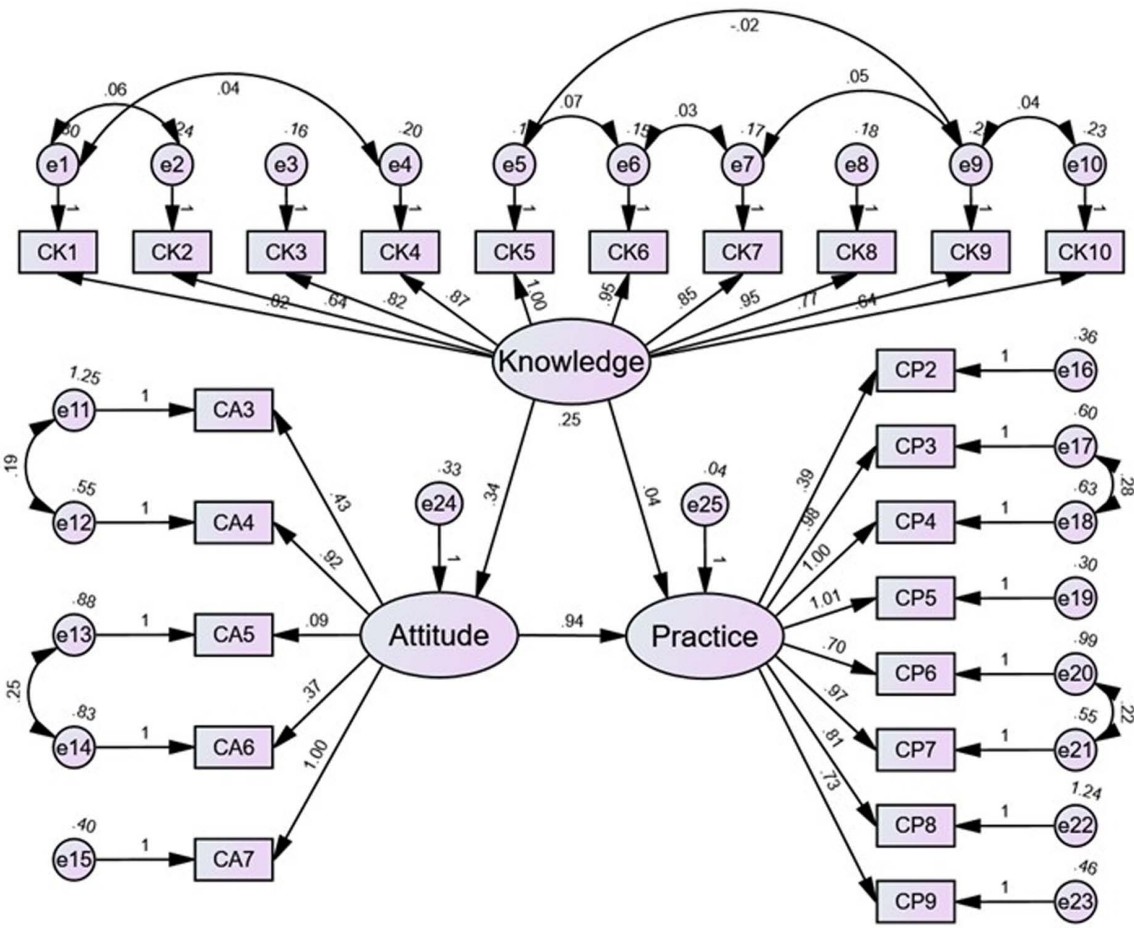

**Fig 2. Structural equation modeling.**

genetics is not part of the traditional Chinese medicine that is followed by many people of older generations. Previous studies showed that younger people have more positive attitudes toward genetic testing than older people [42–44]. Unmarried people were the most knowledgeable about genetic testing in breast cancer, probably because of the fear of carrying a mutation that could influence their marriage prospects and the possibility of transmitting it to their offspring. Those with more pregnancies and deliveries had more positive attitudes toward genetic testing, probably to be aware of the risk of having transmitted it to their children. Urban, higher education, employment, and higher income were associated with higher knowledge and/or attitude scores. It is well-known that the socioeconomic status is a major determinant of health literacy [31]. Patients with luminal B and triple-negative breast cancers had the highest knowledge scores about genetic testing. Since the majority of BRCA1/2-related cancers are triple negative or luminal B [45], it follows the fact that such patients had a higher likelihood of having been talked about genetic testing by their physicians.

This study had some limitations. Firstly, it relied on self-reported data, which may be subject to biases and errors. Secondly, although the minimum sample size was exceeded, the number of participants was unbalanced. Indeed, the high proportion of participants with a low knowledge score could have influenced the results of the multivariable analyses. Thirdly, it was a cross-sectional study, preventing the analysis of causality. A SEM analysis was used to test the

hypotheses based on the KAP theoretical framework (i.e., knowledge influences attitudes and practice, and attitudes influence practice), but such relationships were based on predetermined hypotheses and statistically determined [46]. Further studies with larger sample sizes are still needed.

## Conclusion

In conclusion, this study showed that Chinese patients with breast cancer had poor knowledge, positive attitudes, and suboptimal practices toward genetic testing. More education and counseling on genetic testing for patients are necessary.

## Supporting information

**S1 Table. SEM results.**
(DOCX)

**S2 Table. SEM model fit.**
(DOCX)

## Acknowledgments

We thank the patients who participated in the survey.

## Author contributions

**Conceptualization:** Xin Ye, Jun Cao.

**Data curation:** Xin Ye, Ting Xu, Gang Li, Ming Zhuang, Guangfu Hu, Hongliang Chen, Min Wang, Jie Wang.

**Formal analysis:** Xin Ye, Ting Xu, Min Wang, Jie Wang.

**Methodology:** Xin Ye, Ting Xu, Jun Cao, Ming Zhuang.

**Project administration:** Xin Ye, Ting Xu, Jun Cao.

**Resources:** Gang Li, Hongliang Chen.

**Software:** Guangfu Hu.

**Validation:** Ming Zhuang, Guangfu Hu.

**Visualization:** Gang Li, Hongliang Chen.

**Writing – original draft:** Xin Ye, Ting Xu, Jun Cao, Jie Wang.

**Writing – review & editing:** Xin Ye, Ting Xu, Jun Cao, Jie Wang.

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
