## [Decision Letter · Decision Letter 0]

8 Jan 2025

PONE-D-24-20503Knowledge, attitude, and practice toward genetic testing in breast cancer patientsPLOS ONE

Dear Dr. Ye,

Thank you for submitting your manuscript to PLOS ONE. After careful consideration, we feel that it has merit but does not fully meet PLOS ONE’s publication criteria as it currently stands. Therefore, we invite you to submit a revised version of the manuscript that addresses the points raised during the review process.

We look forward to receiving your revised manuscript.

Kind regards,

Daniele Ugo Tari, M.D.

Academic Editor

PLOS ONE

Journal Requirements:

[This work was supported by the Shanghai Developing Center of Shenkang Hospital (SHDC22020209) and the Shanghai Municipal Key Clinical Specialty (shslczdzk06302)].

Reviewers' comments:

Reviewer's Responses to Questions

**Comments to the Author**

1. Is the manuscript technically sound, and do the data support the conclusions?

Reviewer #1: Yes

Reviewer #2: Partly

2. Has the statistical analysis been performed appropriately and rigorously? 

Reviewer #1: Yes

Reviewer #2: Yes

3. Have the authors made all data underlying the findings in their manuscript fully available?

Reviewer #1: Yes

Reviewer #2: Yes

4. Is the manuscript presented in an intelligible fashion and written in standard English?

Reviewer #1: Yes

Reviewer #2: No

5. Review Comments to the Author

Reviewer #1: This is a cross-sectional study, that aimed to examine for KAP of breast cancer patients regarding genetic testing in China.

There were high proportion of limited knowledge in terms of genetic testing and counselling in this study.

Could you please expain sample size calculation and the expectation of the result.

The lower proportion in terms of limited knowledge score might affect to data implication, which is expected to have in the discussion part.

Many significant subgroups seemed to be absent in the discussion parts, for example, the significant increased knowledge in < 40 YO patients, where the significant altitude scores were observed.

Reviewer #2: In this paper, the authors conduct a cross-sectional study to explore the knowledge, attitudes and practice of breast cancer patients regarding genetic testing in seven public hospitals in Shanghai, China. The paper reports that knowledge, attitude, disease duration of 5-10 years, and triple-negative subtype are among the factors associated with patient behavior regarding genetic testing. More patient education and counseling about genetic testing is needed, the authors conclude.

Despite the fact that this study design has been used widely, the findings reported in this paper – if repeated in a number of robust studies – can be of significance for improving healthcare delivery and clinical decision-making.

With that in mind, this reviewer has the following to remark:

1. Title

The authors could have incorporated China in their title to give the reader an idea of the study population from the start.

2. Introduction

2.1 In the last passage, the authors write, “Therefore, this study aimed to investigate the KAP towards genetic testing in patients with breast cancer.”

The authors are encouraged to be more specific by mentioning China.

2.2 Also, there is a bit of repetition in this section. At times, some editorial attention to improve readability seems necessary.

3. Results

3.1 Another example of the latter include the use of “the.” For instance, in the Results sections, it is mentioned that “The knowledge (r = 0.348, P < 0.001) and attitude (r = 0.464, P < 0.001) were positively related to the practice of patients with breast cancer regarding genetic testing...”

“The” at the beginning of this sentence could have been omitted.

3.2 Table 2 does not seem to be presented in an optimal way. The layout can be improved a great deal.

4. Dicussion

The author note that “A SEM analysis was used to infer causality, but the causality was statistically determined, not observed.”

It should be emphasized that the above statement is not only confusing, but controversial. In general, caution should be exercised in making claims about causality. The use of a structural equation modeling analysis rarely produces results that can be interpreted as causal effects.

Therefore, every sentence about SEM analysis in this paper needs to be rephrased for clarity.

5. Conclusion

The authors end their article stating, “In conclusion, this study showed that patients with breast cancer had poor knowledge, positive attitudes, and suboptimal practices toward genetic testing.”

This statement is subject to misinterpretation. It can be taken as a generalization. As noted above, it is preferable for the authors to stick to the population from which the sample was drawn. They could, for example, say “In conclusion, this study showed that Chinese patients with breast cancer….” or something else that specifies their study population.

I hope this review is helpful and wish the authors the very best with their research!

6. PLOS authors have the option to publish the peer review history of their article (what does this mean? ). If published, this will include your full peer review and any attached files.

**Do you want your identity to be public for this peer review?** For information about this choice, including consent withdrawal, please see our Privacy Policy .

Reviewer #1: No

Reviewer #2: No

---

## [Author Response · Author response to Decision Letter 1]

24 Feb 2025

Title: Knowledge, attitude, and practice toward genetic testing in breast cancer patients

Journal: PLOS ONE

Response to Reviewers’ comments

Dear Editor,

We thank you for your careful consideration of our manuscript. We appreciate your response and overall positive initial feedback, and we have made modifications to improve the manuscript. After carefully reviewing the comments made by the Reviewers, we have modified the manuscript to improve the presentation of our results and their discussion, therefore providing a complete context for the research that may be of interest to your readers.

We hope that you will find the revised paper suitable for publication, and we look forward to contributing to your journal. Please do not hesitate to contact us with other questions or concerns regarding the manuscript.

Best regards,

E-mail: jiewang76@hotmail.com (WJ)

ORCID: 0000-0002-1239-345X 

Reviewer #1

Comment 1: There were high proportion of limited knowledge in terms of genetic testing and counselling in this study.

Response: We agree with the Reviewer. Considering the central and growing place genetic testing is now taking in cancer management, such limited knowledge is troubling. This study highlights the need to improve breast cancer patients’ knowledge of genetic testing.

Comment 2: Could you please expain sample size calculation and the expectation of the result.

Response: We thank the Reviewer. The formula

n=(Z_(1-α/2)/δ)^2×p×(1-p)

was used to calculate the sample size of cross-sectional surveys. In the formula, n represents the sample size for each group, α represents the type I error (which is typically set at 0.05), Z1-α/2=1.96, δ represents the allowable error (typically set at 0.05), and p is set at 0.5 (as setting it at 0.5 maximizes the value and ensures a sufficiently large sample size). Hence, the calculated sample size was 384. Considering an estimated questionnaire response rate of 80%, a minimum of 480 valid questionnaires were needed. It was clarified in the manuscript.

Comment 3: The lower proportion in terms of limited knowledge score might affect to data implication, which is expected to have in the discussion part.

Response: We thank the Reviewer for the comment. Indeed, the high proportion of participants with a low knowledge score could have influenced the results of the multivariable analyses. It was clarified in the Limitations.

Comment 4: Many significant subgroups seemed to be absent in the discussion parts, for example, the significant increased knowledge in < 40 YO patients, where the significant altitude scores were observed.

Response: We thank the Reviewer for the comment. Patients <40 years old could be more likely to know of genetic testing because genetics in oncology is a young discipline with which older patients can be less familiar. Genetics is a topic that has been covered in secondary school in China for a little more than two decades, making younger patients more likely to be knowledgeable about genetics. Young patients are also more likely to access the internet frequently to seek knowledge than older patients. Furthermore, genetics is not part of the traditional Chinese medicine that is followed by many people of older generations. Previous studies showed that younger people have more positive attitudes toward genetic testing than older people [1-3]. Unmarried people were the most knowledgeable about genetic testing in breast cancer, probably because of the fear of carrying a mutation that could influence their marriage prospects and the possibility of transmitting it to their offspring. Those with more pregnancies and deliveries had more positive attitudes toward genetic testing, probably to be aware of the risk of having transmitted it to their children. Urban, higher education, employment, and higher income were associated with higher knowledge and/or attitude scores. It is well-known that the socioeconomic status is a major determinant of health literacy [4]. Patients with luminal B and triple-negative breast cancers had the highest knowledge scores about genetic testing. Since the majority of BRCA1/2-related cancers are triple negative or luminal B [5], it follows the fact that such patients had a higher likelihood of having been talked about genetic testing by their physicians. It was added to the Discussion. 

Reviewer #2

Comment 1: Title

The authors could have incorporated China in their title to give the reader an idea of the study population from the start.

Response: We agree with the Reviewer. The title was revised as “Knowledge, attitude, and practice toward genetic testing in breast cancer patients in China”.

Introduction

Comment 2: In the last passage, the authors write, “Therefore, this study aimed to investigate the KAP towards genetic testing in patients with breast cancer.”

The authors are encouraged to be more specific by mentioning China.

Response: We thank the Reviewer. It was revised as “Therefore, this study aimed to investigate the KAP towards genetic testing in patients with breast cancer in China.”

Comment 3: Also, there is a bit of repetition in this section. At times, some editorial attention to improve readability seems necessary.

Response: We appreciate the Reviewer’s feedback. The Introduction has been proofread to improve clarity and readability.

Results

Comment 4: Another example of the latter include the use of “the.” For instance, in the Results sections, it is mentioned that “The knowledge (r = 0.348, P < 0.001) and attitude (r = 0.464, P < 0.001) were positively related to the practice of patients with breast cancer regarding genetic testing...”

“The” at the beginning of this sentence could have been omitted.

Response: We thank the Reviewer. The manuscript was proofread.

Comment 5: Table 2 does not seem to be presented in an optimal way. The layout can be improved a great deal.

Response: We appreciate the Reviewer’s feedback. The layout of Table 2 has been refined to enhance clarity and readability.

Discussion

Comment 6: The author note that “A SEM analysis was used to infer causality, but the causality was statistically determined, not observed.”

It should be emphasized that the above statement is not only confusing, but controversial. In general, caution should be exercised in making claims about causality. The use of a structural equation modeling analysis rarely produces results that can be interpreted as causal effects.

Therefore, every sentence about SEM analysis in this paper needs to be rephrased for clarity.

Response: We appreciate the Reviewer’s critical insight. The manuscript has been revised to ensure cautious and accurate phrasing regarding the interpretation of SEM analysis results.

Conclusion

Comment 7: The authors end their article stating, “In conclusion, this study showed that patients with breast cancer had poor knowledge, positive attitudes, and suboptimal practices toward genetic testing.”

This statement is subject to misinterpretation. It can be taken as a generalization. As noted above, it is preferable for the authors to stick to the population from which the sample was drawn. They could, for example, say “In conclusion, this study showed that Chinese patients with breast cancer….” or something else that specifies their study population.

Response: Thank you for the suggestion. The Conclusion was revised as suggested: “In conclusion, this study showed that Chinese patients with breast cancer had poor knowledge, positive attitudes, and suboptimal practices toward genetic testing. More education and counseling on genetic testing for patients are necessary.” 

References

1. Makeeva OA, Markova VV, Roses AD, Puzyrev VP. An epidemiologic-based survey of public attitudes towards predictive genetic testing in Russia. Per Med. 2010;7: 291-300. doi:10.2217/pme.10.23

2. Aro AR, Hakonen A, Hietala M, Lonnqvist J, Niemela P, Peltonen L, et al. Acceptance of genetic testing in a general population: age, education and gender differences. Patient Educ Couns. 1997;32: 41-49. doi:10.1016/s0738-3991(97)00061-x

3. Cherkas LF, Harris JM, Levinson E, Spector TD, Prainsack B. A survey of UK public interest in internet-based personal genome testing. PLoS One. 2010;5: e13473. doi:10.1371/journal.pone.0013473

4. Svendsen MT, Bak CK, Sorensen K, Pelikan J, Riddersholm SJ, Skals RK, et al. Associations of health literacy with socioeconomic position, health risk behavior, and health status: a large national population-based survey among Danish adults. BMC Public Health. 2020;20: 565. doi:10.1186/s12889-020-08498-8

5. Sonderstrup IMH, Jensen MR, Ejlertsen B, Eriksen JO, Gerdes AM, Kruse TA, et al. Subtypes in BRCA-mutated breast cancer. Hum Pathol. 2019;84: 192-201. doi:10.1016/j.humpath.2018.10.005

---

## [Decision Letter · Decision Letter 1]

24 Mar 2025

Knowledge, attitude, and practice toward genetic testing in breast cancer patients in China

PONE-D-24-20503R1

Dear Dr. Ye,

We’re pleased to inform you that your manuscript has been judged scientifically suitable for publication and will be formally accepted for publication once it meets all outstanding technical requirements.

Kind regards,

Daniele Ugo Tari, M.D.

Academic Editor

PLOS ONE

Additional Editor Comments (optional):

Dear Authors,

All comments have been addressed.

The paper has been improved, consequently, it can be accepted for publication.

Sincerely,

Reviewers' comments:

Reviewer's Responses to Questions

**Comments to the Author**

1. If the authors have adequately addressed your comments raised in a previous round of review and you feel that this manuscript is now acceptable for publication, you may indicate that here to bypass the “Comments to the Author” section, enter your conflict of interest statement in the “Confidential to Editor” section, and submit your "Accept" recommendation.

Reviewer #1: All comments have been addressed

Reviewer #2: All comments have been addressed

2. Is the manuscript technically sound, and do the data support the conclusions?

Reviewer #1: Yes

Reviewer #2: (No Response)

3. Has the statistical analysis been performed appropriately and rigorously? 

Reviewer #1: Yes

Reviewer #2: (No Response)

4. Have the authors made all data underlying the findings in their manuscript fully available?

Reviewer #1: Yes

Reviewer #2: (No Response)

5. Is the manuscript presented in an intelligible fashion and written in standard English?

Reviewer #1: Yes

Reviewer #2: (No Response)

6. Review Comments to the Author

Reviewer #1: Thank you for your responses to all comments

This could demonstrated results according to authors' hypothesis.

Great work

Reviewer #2: (No Response)

7. PLOS authors have the option to publish the peer review history of their article (what does this mean? ). If published, this will include your full peer review and any attached files.

**Do you want your identity to be public for this peer review?** For information about this choice, including consent withdrawal, please see our Privacy Policy .

Reviewer #1: No

Reviewer #2: No

---

## [Editor Report · Acceptance letter]

PONE-D-24-20503R1

PLOS ONE

Dear Dr. Ye,

I'm pleased to inform you that your manuscript has been deemed suitable for publication in PLOS ONE. Congratulations! Your manuscript is now being handed over to our production team.

Kind regards,

on behalf of

Dr. Daniele Ugo Tari

Academic Editor

PLOS ONE